# Signalment, Immunological and Parasitological Status and Clinicopathological Findings of *Leishmania*-Seropositive Apparently Healthy Dogs

**DOI:** 10.3390/ani13101649

**Published:** 2023-05-16

**Authors:** Marta Baxarias, Oriol Jornet-Rius, Giulia Donato, Cristina Mateu, Mª Magdalena Alcover, Maria Grazia Pennisi, Laia Solano-Gallego

**Affiliations:** 1Departament de Medicina i Cirurgia Animals, Universitat Autònoma de Barcelona, 08193 Bellaterra, Spain; marta.baxarias@uab.cat (M.B.); oriol.jornet@uab.cat (O.J.-R.); 2Dipartimento di Scienze Veterinarie, Università di Messina—Polo Universitario Annunziata, 98168 Messina, Italy; giudonato@unime.it (G.D.); mariagrazia.pennisi@unime.it (M.G.P.); 3Ecuphar Veterinària SLU, 08173 Barcelona, Spain; cmateu@ecuphar.es; 4Departament de Biologia, Sanitat i Medi Ambient, Facultat de Farmacia, Universitat de Barcelona, 08028 Barcelona, Spain; mmagdalenaalcoveramengual@ub.edu

**Keywords:** antibody level, canine leishmaniosis, ELISA, IFN-γ, *Leishmania infantum*, quantitative PCR

## Abstract

**Simple Summary:**

Canine leishmaniosis is a complex infection that ranges from the apparent absence of disease to a severe fatal clinical illness. Numerous epidemiological serosurveys have been carried out in Europe, although these studies often lack assessment of clinical health status. The aim of this study was to evaluate signalment, immunological and parasitological status correlated with clinicopathological findings of *L. infantum*-seropositive apparently healthy dogs living in endemic areas of Europe. Routine laboratory tests, endpoint in-house ELISA to quantify the anti-*Leishmania* antibodies, blood *Leishmania* quantitative PCR (qPCR) and IFN-γ ELISA were performed. Dogs were classified as healthy or sick depending on the results of routine laboratory tests. Seropositive sick dogs presented a worse clinical status that included a higher proportion of medium to high antibody levels and positive qPCR, lower IFN-γ concentration and several clinicopathological findings compared to truly healthy seropositive dogs. In conclusion, further investigation of apparently healthy *L. infantum*-seropositive dogs is necessary in the clinical setting, as a significant proportion of seropositive dogs that are sick with clinicopathological findings remain undetected if routine laboratory tests are not undertaken. Performing these routine laboratory tests with the combination of a thorough physical examination will improve early disease diagnosis, treatment and prognosis of clinical leishmaniosis.

**Abstract:**

Canine leishmaniosis caused by *Leishmania infantum* is a disease with a wide range of clinical manifestations. Epidemiological serosurveys performed in Europe often lack a thorough assessment of clinical health status of studied dogs. The aim of this study was to evaluate signalment, immunological and parasitological status and clinicopathological findings of *L. infantum*-seropositive apparently healthy dogs (n = 212) living in endemic areas. Routine laboratory tests, endpoint in-house ELISA to quantify the anti-*Leishmania* antibodies, blood *Leishmania* qPCR and IFN-γ ELISA were performed. All dogs enrolled were *L. infantum*-seropositive and were classified as healthy (n = 105) or sick (n = 107) according to LeishVet guidelines. The sick group presented a higher proportion of medium to high antibody levels and positive qPCR and lower IFN-γ concentration compared to the healthy group. Sick dogs were mostly classified in LeishVet stage IIa. Biochemical alterations (98%) were the most common clinicopathological findings, with fewer urinary tract (46%) and hematological (40%) alterations. Apparently healthy *L. infantum*-seropositive dogs can be classified between truly healthy dogs and sick dogs with clinicopathological findings. Sick dogs presented medium to high seropositivity and parasitemia and low IFN-γ concentrations, and their most common clinicopathological abnormalities were serum protein alterations followed by proteinuria and lymphopenia.

## 1. Introduction

Canine leishmaniosis (CanL), caused by the protozoan *Leishmania infantum*, is a vector-borne zoonotic disease endemic in the Mediterranean basin [1]. The dog is considered the main domestic reservoir of *L. infantum* infection [2], while other mammals such as wild canids, rodents and lagomorphs might be able to maintain a wild life cycle [3]. *Leishmania* completes its life cycle within two hosts, a phlebotomine sand fly vector, which transmits the promastigote form, and a mammal, where the amastigote form develops [2]. This canine disease is widely variable in clinical manifestations and severity. The most common clinical signs are skin lesions and lymphadenomegaly [2,3,4,5,6,7]. Other common clinical signs are reviewed elsewhere [8]. Diagnosis is achieved through an integrated approach considering signalment, history, clinical findings and results of laboratory tests [2,8]. Laboratory methods used to diagnose CanL can be divided into (1) routine laboratory tests including a complete blood count (CBC), biochemistry profile, urinalysis, serum protein electrophoresis and (2) specific laboratory tests that will aid in the direct (cytology/histology, PCR and parasite culture) or indirect identification of the parasite (anti-*Leishmania* antibodies and parasite-specific cellular immunity markers) [9,10]. Mild to moderate non-regenerative anemia, serum protein imbalances and proteinuria are the most common laboratory abnormalities in dogs with leishmaniosis. In contrast, the prevalence of renal azotemia is low despite the high percentage of dogs with proteinuria, indicating renal involvement [11,12,13].

Development of clinical disease depends on the immune response of the individual host, and two major contrasting patterns are documented: (1) a T cell-mediated protective immune response, where dogs remain infected but they do not progress to the development of clinical illness, and (2) a marked humoral non-protective immune response with reduced or absent T cell-mediated immunity, where dogs develop overt clinical disease [10,14,15]. Therefore, a wide spectrum of clinical illnesses has been described in dogs with leishmaniosis, ranging from mild papular dermatitis due to specific cellular immunity and low humoral responses to more severe clinical presentations characterized by renal damage due to immune complex deposition associated with a massive humoral response and high parasite burden [10].

Some factors such as age, sex, breed, nutrition, host genetics, coinfections and concomitant diseases, immunosuppressive conditions, cytokine environment, parasitic burden, the virulence of the strain, previous infections and method of transmission have been demonstrated to affect the spectrum of the clinical manifestation [16,17,18]. However, the mechanisms for resistance or susceptibility to CanL are not completely understood [10]. Age seems to be an important factor. While some researchers have documented the highest prevalence of leishmaniosis in dogs younger than three years and older than eight years [19], others have documented more advanced stages of the disease in older dogs [4,20]. There is no agreement about the effect of sex on the development of clinical leishmaniosis, although some authors have found an increased prevalence of infections in males [21]. Breed influence has also been documented; crossbreeds, Ibizan Hound [22], Maremma sheepdog [21], Poodle and Yorkshire [23] seem to be less affected by clinical leishmaniosis, while Boxer, German Shepherd, Rottweiler and Cocker Spaniel [5,24] are more susceptible to the disease. Moreover, small breeds and long-coated breeds are at lower risk of developing clinical disease [21].

In endemic regions, where the densities of competent vector and susceptible hosts are high, particularly when there is a lack of preventative measures [25], there is a high prevalence of *L. infantum* infection in apparently healthy dogs [26]. Furthermore, apparently healthy dogs (with no evidence of clinical signs) can be divided into two groups: (1) seropositive dogs and (2) seronegative but PCR-positive dogs [25]. However, the prevalence of clinical illness is frequently lower than 10% [10]. According to longitudinal studies, apparently healthy PCR and antibody-positive dogs living in endemic areas will develop clinical and clinicopathological signs over time [27].

Numerous epidemiological serosurveys in seropositive apparently healthy dogs have been carried out in Europe [6]. However, most of these studies lack a thorough assessment of clinical health status based on a thorough physical examination and routine laboratory tests. Therefore, the aim of this study was to evaluate signalment, immunological and parasitological status combined thoroughly with the clinicopathological findings of seropositive apparently healthy dogs living in endemic areas of *L. infantum* infection.

## 2. Materials and Methods

### 2.1. Dogs

Blood was collected from 212 apparently healthy dogs based on a full physical examination performed between September 2020 and June 2021 by several veterinarians practicing in different areas of Spain (n = 179 dogs) and Italy (n = 33 dogs). Blood samples were collected by jugular or metatarsian venipuncture and transferred immediately into different tubes: ethylenediaminetetraacetic acid (EDTA) tubes for complete blood count (CBC) (Spain: XN1000, Sysmex España SL, Sant Just Desvern, Spain; Italy: Advia 2120, Siemens Healthcare SRL, Milano, Italy) and *Leishmania* quantitative PCR (qPCR); plain serum tubes for serum electrophoresis (Spain and Italy: Capillarys 3, Sebia Dubai SA, Dubai, UAE); biochemistry profile (Spain: Vitros 5600, Ortho Clinical Diagnostics, New York, NY, USA; Italy: AU 5800, Beckman Coulter International SA, Nyon, Switzerland), which included urea, creatinine, total proteins, albumin, total globulins, albumin/globulin (A/G) ratio, alanine aminotransferase (ALT) and alkaline phosphatase (ALP) and the detection of anti-*Leishmania* antibodies; and heparin tubes for IFN-γ release whole blood assay (WBA). Urine samples were collected by free catch or cystocentesis for urinalysis, which included the study of urine specific gravity (USG), sediment analysis, urinary protein/creatinine ratio (UPCR) (Spain: Vitros 5600, Ortho Clinical Diagnostics, New York, NY, USA; Italy: Cobas U601, Roche, Buenos Aires, Argentina) and urine strip test (Beckman Coulter International SA, Nyon, Switzerland) detecting pH, proteins, blood, hemoglobin, acetone, glucose, nitrites, urobilinogen, bilirubin and leukocytes. Hematological and biochemical parameters were considered altered when they were outside the reference intervals. Serum protein electrophoretic patterns were defined following previously published guidelines [28]. Chronic antigenic stimulation was considered when normal to increased total proteins, normal to mild hypoalbuminemia, normal to mild hyperglobulinemia and polyclonal beta and/or gammaglobulinemia were present. When increased alpha-1 or alpha-2 globulins were also detected, an acute phase response with a chronic antigenic stimulation pattern was considered [28].

All apparently healthy dogs enrolled based on a physical examination were seropositive to *L. infantum* and classified into two different groups depending on the results of the routine laboratory tests: (1) seropositive healthy dogs (with the absence of clinicopathological abnormalities) (n = 105) and (2) seropositive sick dogs (with clinicopathological abnormalities) (n = 107). Dogs that presented only values outside the reference intervals due to biological factors such as breed [29], age [30,31,32] or sex [32] were classified as seropositive healthy. Seropositive sick dogs were also classified by LeishVet clinical staging [33].

Furthermore, dogs were also classified by their age for further analyses in (1) young dogs (2 years old or less) and adult dogs (more than 2 years old).

### 2.2. Quantitative ELISA for the Detection of L. infantum-Specific Antibodies

An in-house ELISA was performed on the sera of all dogs studied as previously described [34]. Briefly, samples were diluted to 1:800 in phosphate buffered saline (PBS)-Tween containing 1% dry milk and incubated in *L. infantum* antigen-coated plates (20 µg/mL) for 1 h at 37 °C. Then, the plates were washed three times with PBS-Tween and once with PBS alone and incubated with Protein A conjugated to horseradish peroxidase (concentration 0.16 ng/µL, Peroxidase Conjugate Protein A; Merck KgaA, Darmstadt, Germany) for 1 h at 37 °C. After that, the plates were washed again as described above. The plates were developed by adding the substrate solution *o*-phenylenediamine and substrate buffer (Sigmafast OPD; Merck KgaA, Darmstadt, Germany). The reaction was stopped with 50 µL of 2.5 M H_2_SO_4_. Absorbance values were read at 492 nm by an automatic reader (MB-580 Heales; Shenzhen Huisong Technology Development Co., Ltd., Shenzhen, China). All plates included the serum from a sick dog with confirmed infection as positive control and serum from a healthy dog as a negative control, and all samples were analyzed in duplicate. The result was quantified as ELISA units (EU) related to a positive canine serum used as a calibrator and arbitrarily set at 100 EU. Sera were classified as high positive when having a positivity percentage equal to or higher than 300 EU; medium positive when having a positive percentage equal to or higher than 150 EU and less than 300 EU; and low positive when having a positivity percentage lower than 150 EU and equal to or higher than 35 EU [11].

All samples classified as medium or high positive were further studied using a two-fold serial dilution ELISA. Sera two-fold dilutions were started at 1:800 and continued for 7 to 11 further dilutions. The result was quantified as EU related to a calibrator arbitrary set at 100 EU, with an OD value of 1 at the 1:800 dilution. The mean values of the dilutions at which the optical density (OD) was close to one were chosen for the calculation of the EU using the following formula: (Sample OD/Calibrator OD) × 100 × dilution factor [34].

### 2.3. Blood DNA Extraction and Leishmania quantitative PCR

Blood DNA extraction was performed with a commercial blood DNA extraction kit (MagMAX CORE Nucleic Acid Purification Kit, Thermo Fisher Scientific Inc., Waltham, MA, USA) following the manufacturer’s instructions for a simple workflow with whole EDTA-blood samples, using an automated system (KingFisher Flex Purification System, Thermo Fisher Scientific Inc., Waltham, MA, USA). *Leishmania* qPCR was performed as described elsewhere [35,36,37]. Briefly, qPCR conditions were a two-step temperature (94 and 55 °C) cycling over 45 cycles. Each amplification was performed in a 10 μL reaction mixture containing 1× iTaq supermix with Rox (Bio-Rad), 15 pmol of forward primer (CTTTTCTGGTCCTCCGGGTAGG), 15 pmol reverse primer (CCACCCGGCCCTATTTTACACCAA), 5 pmol labeled TaqMan probe (FAM-TTTTCGCAGAACGCCCCTACCCGC-TAMRA) and 2.5 μL of sample DNA. QuantStudio^TM^ 7Pro (Applied Biosystems, Thermo Fisher Scientific Inc., Waltham, MA, USA) at 95 °C and 55 °C cycling over 45 cycles was used. Amplifications were performed in triplicate for each sample, which included positive and negative controls that were also included in each plate. A 10-fold dilution series of standard DNA from *L. infantum* (CATB101) was used as a calibrator (serial dilution from 10^5^ parasites/mL to 10^−3^ parasites/mL), allowing for the plotting of a standard curve. Results were considered as positive when the quantification cycle (Cq) was <40 and the amplification was detected in all replicates. When only two of the three replicates were detected as positive, qPCR was repeated in the sample to confirm or reject positivity. A total of 147 dogs in the study were analyzed.

### 2.4. IFN-γ Release Whole Blood Stimulation Assay (WBA)

IFN-γ release whole blood stimulation assay (WBA) was performed as described elsewhere [38].

Briefly, three conditions were prepared: medium alone, medium with soluble *L. infantum* antigen (LSA) (1 mg/mL) at a concentration of 10 μg/mL and medium with mitogen concanavalin A (ConA) (100 mg, Medicago AB, Uppsala, Sweden) at a concentration of 10 μg/mL. Afterward, 300 μL of heparinized whole blood was added in each condition. Incubation lasted for 5 days at 37 °C in a 5% CO_2_ environment. Blood was collected in sterile tubes and centrifuged at 300× *g* for 10 min, and supernatants were collected and stored at −80 °C until further use.

IFN-γ was determined in the collected supernatants by a commercial sandwich ELISA following the manufacturer’s instructions (DuoSet^®^ ELISA, R&D Systems, Abingdon, UK). The standard curve for IFN-γ started at 4000 pg/mL, and two-fold dilutions were made until 62.5 pg/mL. Only supernatants with ConA stimulation were diluted with reagent diluent at a proportion of 1:1. Plates were read at 450 nm in a spectrophotometer machine (MB-580 Heales; Shenzhen Huisong Technology Development Co., Shenzhen, China) and processed using a four-parameter logistic curve provided by MyAssays program (http://www.myassays.com/ (accessed on 13 October 2022)). Plates were repeated when the standard curve’s R^2^ value was below 0.98. Samples were analyzed in duplicates. A total of 141 dogs in the study were analyzed. Dogs were classified as IFN-γ producers when *L. infantum* antigen-specific IFN-γ was equal to or higher than 110 pg/mL [39].

### 2.5. Statistical Analysis

A descriptive study of signalment and clinical data of the dogs was performed. Quantitative variables (age, weight, endpoint ELISA and parasite quantification, IFN-γ concentration and numerical clinical data) were assessed using a *t*-test (in normally distributed data) or a Mann–Whitney *U* test (in non-normal distributed data) when two groups were compared (healthy or sick, crossbreed or purebred, male or female and young or adult), while an ANOVA (in normally distributed data) or a Kruskal–Wallis H test (in non-normal distributed data) was used when more than two groups were compared (clinical staging). Qualitative variables (sex, breed, ELISA interpretation, *Leishmania* qPCR interpretation, IFN-γ interpretation and categorical clinical data) were assessed using Fisher’s exact test (when there were two nominal variables and the sample size was small) or a chi-square test (when there were more than two nominal variables and the sample size was big). Spearman’s correlation was also performed to investigate the relation between quantitative variables (age, weight, endpoint ELISA, IFN-γ concentration and numerical clinical data).

A *p*-value  <  0.05 was considered statistically significant. The Shapiro–Wilk test was performed to detect the normal distribution of quantitative variables. The statistical analysis was performed using the package Stats for the software R i386 3.5.1 for Windows.

## 3. Results

### 3.1. Signalment

The quantitative and qualitative characteristics of the dogs are displayed in Table 1. The most common breeds (more than 5 dogs per breed) were Warren Hound (6.6%), American Staffordshire Terrier (3.3%) and German Retriever (3.3%). A total of 41 breeds were included, and the ones not specified had only between 1 to 4 dogs per breed. No differences were found between seropositive healthy dogs and seropositive sick dogs related to breed, sex, age and weight (Table 1).

In-house ELISA results and their interpretation, *Leishmania* qPCR interpretations and IFN-γ concentrations are also displayed in Table 1. There was significantly a higher percentage of low seropositive dogs included in the seropositive healthy group (81%) when compared to the seropositive sick group (30%) (Fisher’s exact test: OR = 10 *p* < 0.0001) (Table 1). Furthermore, the median of EU in endpoint ELISA was significantly lower in seropositive healthy dogs when compared to seropositive sick dogs (Mann–Whitney *U* test: W = 9269 *p* < 0.0001) (Table 1). There was also a lower proportion of *Leishmania* qPCR-positive dogs in the seropositive healthy (28%) when compared to the seropositive sick group (62%) (Fisher’s exact test: OR = 4.1 *p* < 0.0001) (Table 1). Moreover, the median of parasites/mL was significantly lower in seropositive healthy dogs when compared to seropositive sick dogs (Mann–Whitney *U* test: W = 247 *p* = 0.002, Mann–Whitney *U* test: W = 698 *p* = 0.002) (Table 1). There was a higher proportion of IFN-γ producers in the seropositive healthy (58%) when compared to the seropositive sick group (38%) (Fisher’s exact test: OR = 2.3 *p* = 0.026) (Table 1). Furthermore, seropositive healthy dogs also presented a higher IFN-γ concentration in both LSA stimulation (median = 203 pg/mL) and ConA stimulation (median = 6929 pg/mL) when compared to seropositive sick dogs (medians = 20 pg/mL and 5447 pg/mL, respectively) (Mann–Whitney *U* test: W = 1637 *p* = 0.0008; W = 1938 *p* = 0.04) (Table 1).

### 3.2. Clinical Data

The complete blood count (CBC) findings in the different groups are displayed in Table 2. Seropositive sick dogs presented significantly lower red blood cell count (RBC), hemoglobin, hematocrit, mean corpuscular hemoglobin (MCH), mean corpuscular hemoglobin concentration (MCHC) and lymphocytes concentration compared to seropositive healthy dogs (Table 2).

The biochemistry panel, serum electrophoresis and urinalysis findings in the different groups are displayed in Table 3. Seropositive sick dogs presented significantly lower serum albumin, A/G ratio and creatinine when compared to seropositive healthy dogs (Table 3). On the other hand, seropositive sick dogs presented significantly higher total serum proteins, globulins, ALP, alpha-1 globulins, alpha-2 globulins, beta globulins, gamma globulins and UPCR values when compared to seropositive healthy dogs (Table 3).

### 3.3. Clinicopathological Findings in Seropositive Sick Dogs

The distribution of hematologic and biochemical clinicopathological findings in apparently healthy seropositive sick dogs is represented in Table 4. Based on clinicopathological alterations, 107 out of 212 dogs were classified as apparently healthy seropositive sick dogs. A total of 106 each of complete blood counts and biochemistry panels and 88 urinalyses were reviewed.

Forty-two (39.6%) dogs presented at least one hematologic alteration. The most common hematologic clinicopathological findings were lymphopenia (21.7%) followed by leukopenia (7.6%) and anemia (6.6%) (Table 4). Half of the leukopenic dogs (3.8%) had lymphopenic leukopenia, and of the remaining four dogs, three had concurrent neutropenic and lymphopenic leukopenia and one had neutropenic leukopenia (2.8% and 0.9%, respectively) (Table 4). Anemia was mild in all cases, except one that was classified as moderate, and was classified as normocytic and normochromic in all dogs. None of the anemic dogs showed an appropriate regenerative response (reference intervals for reticulocytes < 150.1 × 10^9^/L) [32]. Other less frequent hematologic findings were (frequency less than 5%) eosinophilia (3.7%), mature neutrophilia (4.6%) and altered platelet concentration (2.8% with thrombocytopenia and 1.9% with thrombocytosis) (Table 4).

Almost all dogs (n = 105; 98.1%) presented biochemical abnormalities. Protein alterations were the most common clinicopathological findings in the seropositive sick dogs. Of the dogs with serum protein electrophoresis available (n = 106), the majority of them (82.1%) had serum protein abnormalities. The most common serum protein electrophoretic changes observed were polyclonal hypergammaglobulinemia (74.5%) followed by hyperproteinemia (71.7%), a decreased A/G ratio (64.1%) and hyperglobulinemia (45.3%) (Table 4). Polyclonal hyperbetaglobulinemia alone was present in 24.5% of the dogs, while polyclonal hypergammaglobulinemia in combination with hyperbetaglobulinemia was seen in 17.9% of the dogs. Hypoalbuminemia was only detected in 12 dogs (11.3%). All hypoalbuminemic dogs had increased alpha-2 globulins, and 6 of them had concurrent proteinuria. The most frequent serum protein electrophoretic pattern seen was consistent with chronic antigenic stimulation alone (65.1%) followed by eighteen dogs (17%) with concurrent chronic antigenic stimulation and acute phase response. 

Proteinuria was the second most common clinicopathological finding (Table 4). Proteinuria was present in 35.6% of the seropositive sick dogs in which UPCR was available (n = 87). However, renal azotemia was only present in 1.9% of the dogs. Other biochemical alterations observed were increased hepatic enzymes (13.2%), decreased urea (12.3%) and decreased creatinine (4.7%) concentrations (Table 4).

### 3.4. Relationship between Clinicopathological Findings, Signalment, Antibody Levels, Leishmania Quantitative PCR, IFN-γ Concentration and Clinical Staging

Regarding numerical hematology parameters, the eosinophil concentration was the only parameter to show significant differences when compared with the sex of the dogs. Males showed significantly higher eosinophil concentrations than females (Mann–Whitney *U* test: W = 4417 *p* = 0.022). Regarding age, young dogs tended to have higher total leukocyte, band neutrophil and monocyte concentrations than adult dogs (Mann–Whitney *U* test: W = 2549 *p* = 0.025; W = 3270 *p* = 0.035; W = 2625 *p* = 0.022, respectively). Concerning breed, crossbreed dogs had higher RBC, MCV (*t*-test: t = 2.14, *df* = 194.43, *p* = 0.033; t = −2.73, *df* = 199.28 *p* = 0.0068, respectively) and MCHC values than purebred dogs (Mann–Whitney *U* test: W = 6433 *p* = 0.032). *Leishmania* qPCR-positive dogs showed significantly lower RBC, hematocrit and hemoglobin values (*t*-test: t = 2.68, *df* = 131.19, *p* = 0.008; t = −2.55, *df* = 129.33 *p* = 0.012; t = 2.53, *df* = 124.16, *p* = 0.013, respectively) than qPCR-negative dogs. On the other hand, IFN-γ producer dogs showed significantly higher RBC, hematocrit and hemoglobin values (*t*-test: t = −2.72, *df* = 136.48, *p* = 0.007; t = −3.06, *df* = 138 *p* = 0.003; t = −2.91, *df* = 137.89, *p* = 0.004, respectively) than IFN-γ non-producer dogs. No other differences in hematological parameters were found regarding age, sex, breed, *Leishmania* qPCR, IFN-γ production or clinical staging. When Spearman’s correlation between hematological numerical data and signalment and clinical staging was studied, lower levels of RBC, hematocrit, hemoglobin, reticulocytes, MCV, MCH, lymphocytes and eosinophils were detected with increasing EU, and lower leukocyte and band neutrophil numbers were found with increasing age (Table 5). Higher levels of RBC, hemoglobin, hematocrit and MCHC were detected with increasing IFN-γ concentration in blood stimulated with soluble *L. infantum* antigen, while IFN-γ concentration in blood stimulated with concanavalin A only increased with RBC, hemoglobin and MCHC (Table 5).

When hematologic categorical data were studied, the only hematological parameter significantly different was platelet concentration regarding clinical staging, where thrombocytosis was found with more frequency in LeishVet clinical stage IIa (chi-square: *X*^2^ = 37.76, *df* = 8, *p* < 0.001). No other categorical hematological differences were found regarding signalment or clinical staging.

Regarding numerical biochemical parameters, the only differences found were ALT U/L and UPCR regarding sex and alpha-2 globulin concentration regarding age. Females showed higher ALT levels (Mann–Whitney *U* test: W = 4274 *p* = 0.041), while males had higher UPCR values (Mann–Whitney *U* test: W = 3031.5 *p* = 0.013). Adult dogs had higher values of alpha-2 globulins (Mann–Whitney *U* test: W = 747.5 *p* = 0.045) than young dogs. No statistical differences were noted regarding breed. *Leishmania* qPCR-positive dogs showed several higher values when compared to negative dogs, which included total proteins (Mann-–Whitney *U* test: W = 1531.5 *p* < 0.0001), globulins (Mann–Whitney *U* test: W = 1280 *p* < 0.0001), alpha-1 globulin (Mann–Whitney *U* test: W = 1892 *p* = 0.0023), alpha-2 globulin (Mann–Whitney *U* test: W = 1336.5 *p* < 0.0001), beta globulin (Mann–Whitney *U* test: W = 1892.5 *p* = 0.0024), gamma globulin (Mann–Whitney *U* test: W = 1599.5 *p* < 0.0001) and ALP (Mann–Whitney *U* test: W = 1837.5 *p* = 0.0011). Furthermore, *Leishmania* qPCR-positive dogs showed lower values of albumin (Mann–Whitney *U* test: W = 3672.5 *p* < 0.0001) and A/G ratio (Mann–Whitney *U* test: W = 3610 *p* = 0.0002) when compared to negative dogs. On the other hand, IFN-γ producer dogs showed lower values of globulins (Mann–Whitney *U* test: W = 3016.5 *p* = 0.016), alpha-2 globulin (Mann–Whitney *U* test: W = 2909 *p* = 0.033), gamma globulin (Mann–Whitney *U* test: W = 2954.5 *p* = 0.032) and higher values of the A/G ratio (Mann–Whitney *U* test: W = 1904 *p* = 0.024) compared to IFN-γ non-producer dogs. Regarding clinical staging, stage IIa dogs showed higher concentrations of total proteins and lower ALT levels; stage IIb had lower levels of total proteins, albumin, globulins, gamma globulins, A/G ratio and USG; and stage III showed higher levels of globulins, gamma globulins, UPCR, ALT and urea and had lower levels of albumin, A/G ratio and USG when compared to other stages (Table 6). When Spearman’s correlations between biochemical numerical data and in house ELISA were studied, positive correlations with total proteins, globulins, alpha-2 globulins, beta globulins, gamma globulins, ALP and UPCR and negative correlations with albumin and A/G ratio were found regarding EU. Regarding age, positive correlations with UPCR and ALT and negative with creatinine concentrations were found (Table 7). A positive correlation was also observed between IFN-γ concentration (in both blood stimulated with LSA and with ConA) and albumin and A/G ratio (Table 7). On the other hand, a negative correlation was observed between IFN-γ concentration (in both blood stimulated with LSA and with ConA) and total proteins, globulins, alpha-2 globulins and gamma globulins (Table 7).

When biochemical categorical alterations were studied, males showed a higher frequency of increased alpha-2 globulins (chi-square: *X*^2^ = 3.99; *df* = 1, *p* = 0.046) and a decreased A/G ratio (chi-square: *X*^2^ = 4.87; *df* = 1, *p* = 0.027). Regarding the clinical staging, serum protein alterations were found in all stages; however, a higher degree of hyperglobulinemia and hypergammaglobulinemia, decreased A/G ratio, increased alpha-2 globulins and renal alterations (isosthenuria, proteinuria and renal azotemia) was observed with more advanced clinical stages (IIb and III) (Table 8). In addition, thrombocytosis and serum electrophoresis pattern corresponding to chronic antigenic stimulation was more frequently observed in clinical stage IIa (Table 8).

## 4. Discussion

In endemic regions such as Spain and Italy, a high prevalence of *L. infantum* infection in apparently healthy dogs exists [26], while clinical leishmaniosis is usually developed by a limited proportion of the infected dogs [10,44]. According to a longitudinal study, apparently healthy but *L. infantum*-seropositive dogs will develop clinical and clinicopathological signs over time [27]. However, these apparently healthy *L. infantum*-seropositive dogs are fairly unknown and usually neglected in the clinical setting, and few recommendations have been published regarding their monitoring and treatment [33,45]. In recent years and in endemic areas, due to an increased awareness of the disease and the use of advanced laboratory diagnostic tests that allow earlier diagnosis in the clinical setting, apparently healthy *L. infantum*-seropositive sick dogs without overt clinical signs but with the presence of laboratory abnormalities usually associated with leishmaniosis are a frequent finding. Therefore, this study describes the signalment and clinical data of apparently healthy *L. infantum*-seropositive dogs and the most common clinicopathological findings in the population of apparently healthy *L. infantum*-seropositive sick dogs without evident clinical signs.

In the present study, an important difference regarding serological status was found between *L. infantum*-seropositive healthy dogs (with subclinical infection and no clinical illness) and *L. infantum*-seropositive sick dogs. Sick seropositive dogs presented higher median endpoint EUs than healthy seropositive dogs and a higher proportion of dogs with high to medium antibody levels. These results are in concordance with previous studies that reported that dogs with high antibody levels are usually sicker and show more pronounced clinicopathological abnormalities and, thus, are classified in higher clinical stages of leishmaniosis [34,46,47,48]. In addition, an increase in endpoint EUs was correlated to several clinical parameters that indicated a clinical worsening such as an increase in total proteins, globulins and UPCR and a decrease in the A/G ratio, RBC and hematocrit.

Interestingly, the risk of seropositivity to *L. infantum* is associated with several factors related inherently to the dog such as age, breed and the dog’s environment, among others. The risk of seropositivity to *L. infantum* has been reported to increase with the dog’s age, which seems to be related to repeated exposure to *Leishmania* [21,49,50,51], although a bimodal age distribution with one peak in young dogs (under 2 years old) and a second peak in older dogs (over 8 years old) has also been commonly reported [52]. Sex has also been documented to be a risk factor for seropositivity to *L. infantum,* with male dogs presenting a higher risk of exposure to *Leishmania* infection than female dogs [21,51], although other studies did not detect differences between male and female dogs [49,53]. However, sex could be associated with other factors that could increase the probability of seropositivity to *L. infantum* such as the size of the dog, being used as a guard dog and living outdoors [26,54]. Environmental factors such as living outdoors or indoors have also been identified as an important risk factor for seropositivity to *L. infantum* [21,49,52,53]. In this study, no differences in age, sex, breed and weight were observed between *L. infantum*-seropositive healthy dogs and sick dogs, which could indicate that even if the characteristics of the dogs could be a risk factor for being *L. infantum*-seropositive, it does not seem to present a risk factor to progression towards disease development, worsening of clinicopathological abnormalities and ELISA results. However, several correlations with age and clinical data (ALT, UPCR and creatinine levels) were observed and could be easily explained by biological factors, as older dogs tend to present clinicopathological abnormalities due to age-related diseases [30], which could be a risk factor and affect disease development and worsening.

Moreover, *L. infantum*-seropositive sick dogs also presented a higher proportion of *Leishmania* qPCR-positive dogs, which indicates a higher parasitemia and parasite dissemination and, thus, a worse situation and a higher probability to develop clinical signs than seropositive healthy dogs [2,34]. In the present study, positivity to *Leishmania* qPCR was also correlated to several parameters that could indicate a clinical worsening such as higher concentrations in total proteins and globulins and lower concentrations in RBC, hematocrit, and hemoglobin.

Furthermore, *L. infantum*-seropositive sick dogs not only presented higher antibody levels and PCR positivity but also presented a significantly lower IFN-γ production in both blood stimulated with LSA and blood stimulated with ConA when compared to *L. infantum*-seropositive healthy dogs. Additionally, a higher proportion of IFN-γ producers were observed in the seropositive healthy group. Previous studies have reported that a lower IFN-γ concentration can often be correlated to higher antibody levels and blood parasitemia and, thus, more severe clinical stages of CanL [38,55,56]. In addition, in the present study, IFN-γ concentration was also positively correlated to several values such as RBC, hemoglobin, hematocrit, albumin and the A/G ratio, while it was negatively correlated to total proteins and globulins. These correlations could indicate a lower probability of clinical worsening if the dog presents IFN-γ production. For example, in a previous study performed in dogs with clinical leishmaniosis during treatment [38], dogs were classified at the diagnosis between IFN-γ producers and IFN-γ non-producers. IFN-γ non-producer dogs were also classified in more severe stages than IFN-γ producer dogs similarly to the present study. Moreover, IFN-γ non-producer dogs presented a significant increase in IFN-γ production and clinical improvement during treatment [38]. In the present study, IFN-γ non-producer dogs presented higher globulins, alpha-2 globulin and gamma globulin values and lower A/G ratio, RBC, hematocrit and hemoglobin values, which indicate a worse clinical situation than that of IFN-γ producer dogs.

Regarding clinical stages, the LeishVet group proposed a classification of four clinical stages (from mild disease in stage I to very severe disease in stage IV) based on clinical signs, clinicopathological abnormalities and serological status [33]. This tool also suggests different treatment protocols and prognoses for each clinical stage and can be used in the clinical setting. In a study in Spain [34], a group of dogs was diagnosed with leishmaniosis, classified by LeishVet stages and followed-up during treatment. The majority of these dogs (86%) were classified before treatment as stage II, and most of them (75%) were further sub-classified as stage IIa, while stage IIb presented a lower proportion (25%) [34]. Furthermore, few dogs (14%) were classified as stage III [34]. Similarly, in the present study, only apparently healthy *L. infantum*-seropositive sick dogs were classified by LeishVet clinical stages, and most of them were classified as stage IIa (55.1%), while stage IV had the lowest proportion of dogs (0.9%). These results were to be expected, as the dogs did not present any clinical signs and most of them presented few clinicopathological abnormalities, which is more common in lower stages of CanL (IIa and IIb) [33]. On the other hand, seropositive healthy dogs were not further classified, as none of them presented clinical signs or clinicopathological abnormalities.

Regarding clinical data of seropositive sick dogs, in concordance with previous reports, the most frequent clinicopathological alteration in our study was dysproteinemia usually characterized by the presence of hyperproteinemia secondary to hyperglobulinemia, specifically due to an increase in the gamma globulin or beta globulin proteins and less frequently alpha-2 globulins and a decreased A/G ratio [5,7,11,13,20,57,58,59,60]. In addition, our study showed that the most frequent serum electrophoretic pattern observed in the seropositive sick group was consistent with chronic antigenic stimulation. These findings are related to the exaggerated humoral response with a polyclonal proliferation of B-lymphocytes and the consequent production of non-protective anti-*Leishmania* antibodies seen in those diseased dogs with leishmaniosis [6,25,61]. Moreover, in contrast with other studies, the frequency of hypoalbuminemia was low (11.3%) and mostly associated with increased alpha-2 globulins, suggesting a probable ongoing active inflammation [5,58]. However, hypoalbuminemia related to or exacerbated by the presence of proteinuria should also be considered. Additionally, the low frequency of liver and renal involvement observed in our study could also influence the lesser degree of hypoalbuminemia [5,58]. In addition, as expected, since an uncontrolled humoral response will reflect on protein concentrations, positive correlations between total proteins, globulins, alpha-2 globulins, beta and gamma globulins and EU and a negative correlation between albumin and A/G ratio and EU were found.

Renal disorders are also a frequent feature in dogs diagnosed with leishmaniosis [58,62,63,64], with renal azotemia and proteinuria being the most common laboratory abnormalities indicating renal involvement. Although a high prevalence of renal pathology is detected by histopathology [62,65,66], routine renal parameter alterations stating renal compromise are less frequently observed [4,5,11,20,63]. The kidney disease associated with CanL is primarily of glomerular origin as a consequence of the deposition of circulating immune complexes at different levels of the glomeruli structure [62,66]. Initially, renal involvement is observed by the presence of proteinuria without azotemia. As glomerular damage progresses, secondary tubulointerstitial nephritis and azotemia develop, leading to end-stage renal failure or nephrotic syndrome, the most striking cause of death in CanL [58,62,63,66]. Our study, in agreement with previous reports, shows renal azotemia as a rare clinicopathological finding, while proteinuria without renal azotemia was the second most common laboratory abnormality in seropositive sick dogs at the time of diagnosis [11,20]. We also found a great proportion of dogs with inadequate USG; however, only a few were between isosthenuria levels, and other causes of polyuria/polydipsia were not ruled out. Therefore, renal involvement was not confirmed in those seropositive sick dogs that presented with inadequate USG as the sole altered renal parameter. When the relationship between signalment, EU and biochemical parameters was studied, a positive correlation was observed between UPCR and antibody levels as previously reported in serum [11] and urine samples [67,68]. This result is expected since proteinuria is caused by immune-mediated glomerulonephritis [69]. Moreover, we found a positive correlation between age and UPCR, most likely explained by a more deteriorated renal function in older dogs in addition to the fact that older dogs tend to have more renal and hematologic alterations [20], and a negative correlation between creatinine levels and age, probably associated with decreased muscle mass in older dogs.

Leukogram changes are considered infrequent and have shown a great variability between previously published studies [4,7,11,70]. Our results agree with other studies, where a normal leukogram pattern is the most common observation [4,11,13,70]. Lymphopenia alone or with concurrent mild leukopenia or neutrophilia was the second most common leukogram change and the third more common clinicopathological alteration. These leukogram changes suggest a stress response due to increased endogenous glucocorticoids, usually present in sick animals [7,11,71]. Since other less frequent leukogram changes were also detected, indicating a multifactorial origin of these alterations (enhanced recruitment in several organs and decreased production due to high bone marrow parasitism and inflammation), an individual evaluation of the leukogram changes in dogs with leishmaniosis is recommended to determine the principal ongoing pathogenic mechanism. In addition, we found a negative correlation between lymphocyte, monocyte and eosinophil concentrations and EU. As published before, these results could be related to a bone marrow dysfunction associated with higher parasitism [12,72]. However, the lack of concurrent cytopenias at the same time, the clinical significance of eosinopenia and monocytopenia and decreased lymphocyte numbers with increasing EU could be related to a stress response or enhancement migration of lymphocytes to targeted organs due to *L. infantum* infection in dogs with more severe clinicopathological alterations. A negative correlation between total leukocytes and band neutrophil concentrations with age was also observed; nevertheless, this finding may lack of clinical significance since young dogs tend to have increased white blood cells and band neutrophils numbers [73].

Mild to moderate non-regenerative anemia is a common laboratory finding in dogs with leishmaniosis [5,7,11,13,20,58,70]. Although multifactorial, decreased erythropoiesis due to chronic inflammation is thought to be an important pathogenic mechanism leading to anemia of chronic disease [12,74]. Other described factors involved in the pathogenesis of anemia in CanL are renal disease, chronic bleeding (epistaxis, skin lesion and gastrointestinal ulceration), myelodysplastic syndrome, decreased lipid fluidity of the erythrocyte membrane and, much less likely, the production of anti-erythrocyte antibodies—an immune-mediated mechanism [58,75,76,77,78]. In the present study, the frequency of anemic patients was low and anemia was classified as mild to moderate normocytic/normochromic non-regenerative in all anemic dogs. The prevalence of anemia in the seropositive sick dogs group was low (6.6%) in contrast with previous studies where anemia ranged between 40% to 70% [5,11,20,58,70]. However, our study is in agreement with others where anemia was a less frequent finding in those subclinical *L. infantum*-seropositive sick dogs than in dogs showing overt clinical signs [4,13,60,74]. We also found a negative correlation between EU and erythrogram-related parameters including RBC, hematocrit, hemoglobin, reticulocytes and MCV. These findings suggest that apparently healthy *L. infantum*-seropositive sick dogs present more severe clinicopathological findings with increasing antibody levels as previously reported [55].

Hemostatic disorders such as epistaxis, hematuria and hemorrhagic diarrhea have been reported in CanL [4,5,7,11,58]. Furthermore, these clinical signs have been associated with primary homeostasis defects (thrombocytopathy or vasculitis), and mucosal ulcerative lesions and appear to be unrelated to decreased platelet concentration [59]. In the present study, the frequency of thrombocytopenia was low (2.8%), in agreement with previous studies [11,13,58,70] but in contrast with other studies where the frequency ranged between 20 and 50% [79]. These controversial results could be explained, as the platelet concentration can be reduced in those dogs with overt leishmaniosis where renal disease, bone marrow dysfunction and inflammation are more frequently seen [57,80]. Additionally, thrombocytosis was infrequent in seropositive sick dogs (1.9%) as reported previously [11].

Finally, hepatocyte damage was uncommon, in agreement with other studies [4,5,11,57,58]. Furthermore, regarding liver parameters, a positive relationship was found between age and ALT levels and between EU and ALP levels. ALT is a nonspecific marker of hepatocyte damage that could be increased with a numerous group of diseases frequently found in older dogs [28]. Thus, this observation could be the explanation for the trend of higher levels of ALT in older dogs. ALP is a marker of cholestasis in dogs. In addition, ALP can be affected by endogenous or exogenous cortisol levels [28]. Therefore, the observation of a stress response in our seropositive sick dogs could explain the relationship between ALP levels and EU.

It is important to highlight that there is little information on the monitoring, treatment and prognosis of apparently healthy *L. infantum*-seropositive dogs both in vitro and in vivo therapeutic potential treatments [2,10,45,81]. Published recommendations on how to treat these dogs include the use of repellents, the performance of a follow-up without treatment or a short treatment with conventional anti-*Leishmania* drugs [2,45]. Treatment with immunotherapeutic drugs could also improve the immune response of the dog and avoid further disease development [82]. In this study, we observed that some of the apparently healthy *L. infantum*-seropositive dogs present some clinicopathological alterations that could be undetected if routine laboratory tests are not performed. These results highlight the importance of performing routine laboratory tests in all dogs that are seropositive to *L. infantum*, even when apparently healthy. The performance of routine laboratory tests in these cases should be able to detect early disease development, shorten the treatment and improve the prognosis of clinical leishmaniosis, preventing the spread of the diseases that concern public health.

## 5. Conclusions

Apparently healthy *L. infantum*-seropositive dogs can be further classified into truly healthy dogs and sick dogs with clinicopathological abnormalities. Seropositive healthy dogs tend to present low seropositivity and parasitemia and have a high IFN-γ concentration, while sick dogs with clinicopathological abnormalities tend to present medium to high seropositivity and parasitemia and have a lower IFN-γ concentration. Furthermore, more than half of apparently healthy *L. infantum*-seropositive sick dogs were classified as LeishVet stage IIa, and their most common clinicopathological abnormalities were serum protein alterations followed by proteinuria and lymphopenia.

## Figures and Tables

**Table 1 animals-13-01649-t001:** Qualitative and quantitative clinical characteristics of the seropositive healthy and sick dogs.

Qualitative Characteristics		Total (n = 212) % (95% CI)	Seropositive Healthy (n = 105) % (95% CI)	Seropositive Sick (n = 107) % (95% CI)	*p*-Value (Fisher’s Exact Test)
**Breed**	**Crossbreed**	43.9 (37.1–50.8)	46.7 (36.9–56.7)	41.1 (31.7–51)	0.49
	**Purebred**	56.1 (49.2–62.9)	53.3 (43.3–63.1)	58.9 (48.9–68.3)	
**Sex**	**Female**	43.4 (36.6–50.4)	41 (31.5–51)	45.8 (36.1–55.7)	0.49
	**Male**	56.6 (49.6–63.4)	59 (49–68.5)	54.2 (44.3–63.9)	
**ELISA positivity**	**High or medium**	44.8 (38–51.8)	19 (12–27.9)	70.1 (60.5–78.6)	<0.0001
	**Low**	55.2 (48.2–62)	81 (72.1–88)	29.9 (21.4–39.5)	
***Leishmania*** **qPCR (n = 147)**	**Positive**	44.9 (36.7–53.3)	28.4 (18.5–40.1)	61.6 (49.5–72.8)	<0.0001
	**Negative**	55.1 (46.7–63.3)	71.6 (60–81.5)	38.4 (27.2–50.5)	
**LSA IFN-γ production (n = 141)**	**Producer**	46.8 (38.4–55.4)	58.3 (44.9–70.9)	38.3 (27.7–49.7)	0.026
	**Non-producer**	53.2 (44.6–61.6)	41.7 (29.1–55.1)	61.7 (50.3–72.3)	
**LeishVet stage ***	**II ****	NA	NA	19.6 (12.6–28.4)	NA
	**IIa**			55.1 (45.2–64.8)	
	**IIb**			9.4 (4.6–16.5)	
	**III**			15 (8.8–23.1)	
	**IV**			0.9 (0–5.1)	
**Quantitative Characteristics**		**Total (n = 212)** **Median (min–max)**	**Seropositive Healthy (n = 105)** **Median (min–max)**	**Seropositive Sick (n = 107)** **Median (min–max)**	** *p* ** **-Value** **(Mann-Whitney *U* test)**
**Age (years)**		5 (0.5–14)	4 (1–14)	5 (0.5–12)	0.09
**Weight (kg)**		22 (3–62)	23 (6–62)	20 (3–58)	0.45
**Endpoint ELISA (EU)**		247 (51–61286)	137 (51–1181)	789 (75–61286)	<0.0001
***Leishmania*** **qPCR (parasites/mL) (n = 147)**		0.1 (0.003–886.6)	0.01 (0.004–5.86)	0.48 (0.003–886.6)	0.002
**IFN-γ concentration (pg/mL) (n = 141)**	**Medium with LSA**	80 (0–14,190)	203 (0–14,190)	20 (0–8001)	0.0008
	**Medium with ConA**	5985 (805–43,290)	6929 (1147–43,290)	5447 (805–38,198)	0.04

* Only sick dogs can be classified in LeishVet staging. ** Some dogs (n = 21) could not be classified as stage IIa or more advanced stages due to lack of urinalysis and urinary protein creatinine ratio. Abbreviations: CI: confidence interval, ConA: concanavalin A, EU: ELISA units, max: maximum, min: minimum, qPCR: quantitative PCR, LSA: *Leishmania* soluble antigen, NA: not applicable.

**Table 2 animals-13-01649-t002:** Complete blood count (CBC) parameters of the seropositive healthy and sick dogs.

CBC Parameters (Units)	Reference Intervals * [32,40]	Total (n = 212) Median (Min–Max)	Seropositive Healthy (n = 105) Median (Min–Max)	Seropositive Sick (n = 107) Median (Min–Max)	*p*-Value
**RBC (10^6^/µL)**	5.1–7.6	6.6 (4–9.1)	6.8 (5–9.1)	6.1 (4–8.3)	<0.0001 ^a^
**Hemoglobin (g/dL)**	12.4–19.2	16.2 (10.4–22.5)	17.1 (13.4–22.5)	15.2 (10.4–19.8)	<0.0001 ^b^
**Hematocrit (%)**	35–52	47 (29–62)	49 (36–62)	44 (29–59)	<0.0001 ^c^
**MCV (fL)**	60–77	71 (59–81)	71 (59–80)	71 (61–81)	0.27
**MCH (pg)**	21.9–26.3	24.6 (20.2–33.3)	24.9 (20.2–33.3)	24.2 (21.5–28.6)	0.02 ^d^
**MCHC (g/dL)**	34.4–38.1	34.4 (29.5–45.8)	35.2 (29.9–45.8)	34 (29.5–39.7)	<0.0001 ^e^
**WBC (10^9^/L)**	5.6–20.4	9.5 (3.4–23.5)	9.4 (4.8–23.5)	9.6 (3.4–22.7)	0.91
**Neutrophils (10^9^/L)**	2.9–13.6	6.1 (2.4–20)	6 (2.7–17.2)	6.2 (2.4–20)	0.52
**Lymphocytes (10^9^/L)**	1.1–5.3	1.9 (0.2–4.7)	2.1 (0.4–4.5)	1.7 (0.2–4.7)	<0.0001 ^f^
**Monocytes (10^9^/L)**	0.4–1.6	0.4 (0–2)	0.4 (0.1–1.6)	0.4 (0–2)	0.49
**Eosinophils (10^9^/L)**	0.1–3.1	0.4 (0–4)	0.5 (0–4)	0.4 (0–3.3)	0.47
**Platelets (10^3^/µL) ****	200–500	Adequate	Adequate	Adequate	-

^a^ *t*-test: t = −6.1; ^b^ *t*-test: t = −6.8; ^c^ *t*-test: t = −5.9; ^d^ Mann–Whitney *U* test: W = 4566; ^e^ Mann–Whitney *U* test: W = 4062; ^f^ Mann–Whitney *U* test: W = 4066. * Reticulocytes (reference interval < 150.1 × 10^9^/L) and basophils (reference interval 0–200/µL) are not included in the table due to low numbers and non-significance. ** The majority of platelet concentration results were obtained qualitatively due to platelet aggregation. Abbreviations: CBC: complete blood count, max: maximum, MCH: mean corpuscular hemoglobin, MCHC: mean corpuscular hemoglobin concentration, MCV: mean corpuscular volume, min: minimum, RBC: red blood cells concentration, WBC: leukocytes concentration.

**Table 3 animals-13-01649-t003:** Biochemistry panel, serum electrophoresis and urinalysis parameters of the dogs.

Parameters (Units)	Reference Intervals [41,42,43]	Total (n = 212) Median (Min–Max)	Seropositive Healthy (n = 105) Median (Min–Max)	Seropositive Sick (N = 107) Median (Min–Max)	*p*-Value
**Total protein (g/L)**	54–71	71 (54–117)	67 (54–83)	75 (59–117)	<0.0001 ^a^
**Albumin (g/L)**	26–33	33 (24–56)	34 (26–52)	33 (24–56)	0.002 ^b^
**Globulin (g/L)**	27–44	37 (26–81)	32 (26–41)	42 (29–81)	<0.0001 ^c^
**A/G ratio**	0.86–1.93	0.9 (0.4–1.7)	1.1 (0.7–1.7)	0.8 (0.4–1.5)	<0.0001 ^d^
**ALT (U/L)**	21–102	43 (12–1037)	44 (18–132)	42 (12–1037)	0.19
**ALP (U/L)**	20–156	47 (14–1271)	44 (14–208)	55 (20–1271)	0.001 ^e^
**Creatinine (mg/dL)**	0.5–1.5	0.9 (0.4–3.9)	0.9 (0.5–1.4)	0.8 (0.4–3.9)	0.01 ^f^
**Urea (mg/dL)**	21.4–59.9	35 (14–155)	35 (14–77)	34 (14–155)	0.81
**Serum electrophoresis (g/L)**					
**Albumin**	24.4–49.6	34.5 (19.1–49.1)	36.7 (25.5–45.4)	31.5 (19.1–49.1)	<0.0001 ^g^
**Alpha-1 globulin**	1.7–4.5	3.5 (1.7–8.4)	3.4 (1.7–4.8)	3.7 (2–8.4)	0.008 ^h^
**Alpha-2 globulin**	3.8–10.2	7.3 (2.9–18.2)	6.5 (2.9–12.7)	8.1 (4.3–18.2)	<0.0001 ^i^
**Beta globulin**	8–18	13.2 (7.5–37.9)	12.3 (7.5–17.9)	14.7 (9.2–37.9)	<0.0001 ^j^
**Gamma globulin**	2.6–11.7	10.1 (4.4–60.6)	7.9 (4.4–11.9)	15.1 (5.3–60.6)	<0.0001 ^k^
**UPCR**	<0.5	0.1 (0–101.8)	0.1 (0.02–0.44)	0.2 (0–101.8)	0.007 ^l^
**USG (g/L)**	1006–1050	1034 (1007–1058)	1036 (1016–1058)	1031 (1007–1056)	0.06

^a^ Mann–Whitney *U* test: W = 9144; ^b^ Mann–Whitney *U* test: W = 4252; ^c^ Mann–Whitney *U* test: W = 10214; ^d^
*t*-test: t = −14.2; ^e^ Mann–Whitney *U* test: W = 4656; ^f^ Mann–Whitney *U* test: W = 4466; ^g^
*t*-test: t = −7.7; ^h^ Mann–Whitney *U* test: W = 6673; ^i^ Mann–Whitney *U* test: W = 8060; ^j^ Mann–Whitney *U* test: W = 8188; ^k^ Mann–Whitney *U* test: W = 10126; ^l^ Mann–Whitney *U* test: W = 4913. Abbreviations: A/G: albumin/globulin, ALP: alkaline phosphatase, ALT: alanine transaminase, max: maximum, min: minimum, UPCR: urinary protein creatinine ratio, USG: urine specific gravity.

**Table 4 animals-13-01649-t004:** Distribution of the most common clinicopathological findings in seropositive sick dogs.

Clinicopathological Findings	Number of Dogs (%; 95% CI)
**Hematological alterations (n = 106)**	**42 (39.6;30.3–49.6)**
Anemia	7 (6.6;2.7–13.3)
Lymphopenia	23 (21.7;14.3–30.8)
Leukopenia	8 (7.6;3.3–14.3)
Lymphopenic leukopenia	4 (3.8;1–9.4)
Neutropenic leukopenia	1 (0.9;0.02–5.1)
Neutropenic and lymphopenic leukopenia	3 (2.8;0.6–8.1)
Neutrophilic leukocytosis	1 (0.9;0.02–5.1)
Neutrophilia	3 (2.8;0.6–8.1)
Neutrophilia and lymphopenia	1 (0.9;0.02–5.1)
Eosinophilia	3 (2.8;0.6–8.1)
Eosinophilia and monocytosis	1 (0.9;0.02–5.1)
Thrombocytopenia	3 (2.8;0.6–8.1)
Thrombocytosis	2 (1.9;0.2–6.7)
**Biochemical alterations (n = 107)**	**105 (98.1;93.4–99.8)**
**Serum protein alterations (n = 106)**	**87 (82.1;73.4–88.9)**
Hyperproteinemia	76 (71.7;62.1–80)
Hypoalbuminemia	12 (11.3;6–18.9)
Hyperglobulinemia	48 (45.3;35.6–55.3)
Decreased A/G ratio	68 (64.1;54.3–73.2)
Polyclonal hypergammaglobulinemia	79 (74.5;65.1–82.5)
Polyclonal hyperbetaglobulinemia	26 (24.5;16.7–33.8)
Polyclonal hypergammaglobulinemia and hyperbetaglobulinemia	18 (17.9;11.1–26.6)
**Protein electrophoretic patterns (n = 106)**	**87 (82.1;73.4–88.9)**
Chronic antigenic stimulation	69 (65.1;55.2–74.1)
Chronic antigenic stimulation and acute phase response	18 (17;10.4–25.5)
Normal	19 (17.9;11.2–26.6)
**Renal alterations**	**40 (46;35.2–57)**
Proteinuria (n = 87)	31 (35.6;25.7–46.6) ^a^
Inadequate USG	40 (46;35.2–57)
Isosthenuria (n = 87)	10 (11.5;5.7–20.1) ^a^
Renal azotemia (n = 106)	2 (1.9;0.2–6.7)
**Increased hepatic enzymes (n = 106)**	**14 (13.2;7.4–21.2)**
**Others (n = 106)**	**23 (21.7;14.3–30.8)**
Low urea	13 (12.3;6.7–20.1)
High urea	6 (5.7;2.1–11.9) ^a^
Low creatinine	5 (4.7;1.6–10.7)
High creatinine	3 (2.8;0.6–8.1) ^a^

Abbreviation: A/G: albumin/globulin, CI: confidence interval. ^a^ Two of these dogs were interpreted as having renal azotemia.

**Table 5 animals-13-01649-t005:** Spearman’s correlation (*rs*) between hematological parameters and age, serology and IFN-γ concentration after LSA or ConA stimulations.

CBC Parameters	Age		Serology ELISA Units		IFN-γ Concentration (LSA Stimulation)		IFN-γ Concentration (ConA Stimulation)	
	** *r_s_* **	** *p-* ** **Value**	** *r_s_* **	** *p-* ** **Value**	** *r_s_* **	** *p-* ** **Value**	** *r_s_* **	** *p-* ** **Value**
**RBC**	0.01947	0.7786	−0.3382	**<0.0001**	0.2742	**0.001**	0.1746	**0.0391**
**Hemoglobin**	0.08224	0.2342	−0.3831	**<0.0001**	0.3046	**0.0003**	0.2039	**0.0157**
**Hematocrit**	0.0547	0.4293	−0.4288	**<0.0001**	0.2801	**0.0008**	0.09803	0.2492
**Reticulocytes**	−0.02686	0.8332	−0.5704	**<0.0001**	−0.1212	0.5816	0.02174	0.9216
**MCV**	0.0464	0.5026	−0.1664	**0.0155**	−0.07767	0.3617	−0.1441	0.0894
**MCH**	0.08829	0.2015	−0.2123	**0.0019**	0.1062	0.2118	0.07653	0.3688
**MCHC**	0.05852	0.3977	−0.07718	0.2643	0.2278	**0.0068**	0.2848	**0.0006**
**WBC**	−0.1429	**0.0390**	−0.08484	0.222	0.07476	0.3818	0.02388	0.7803
**Neutrophils conc**	−0.09672	0.1646	−0.05535	0.4272	0.04511	0.5993	−0.04691	0.5849
**Band Neutrophils conc**	−0.1475	**0.0322**	0.0853	0.2172	0.00141	0.9868	0.00279	0.9739
**Lymphocytes conc**	−0.1089	0.1174	−0.161	**0.0202**	0.08133	0.343	0.1589	0.0626
**Monocytes conc**	−0.1048	0.1291	0.1221	0.0767	−0.04173	0.6245	0.03668	0.667
**Eosinophils conc**	−0.1263	0.0677	−0.1483	**0.0317**	0.063	0.4596	0.03614	0.6717
**Basophils conc**	−0.07155	0.3009	−0.2184	**0.0014**	0.0234	0.7837	−0.00702	0.9344
**Platelets conc ***	-	-	-	-	-	-	-	-

Abbreviations: CBC: complete blood count, ConA: concavalin A, conc: concentration, LSA: soluble *L. infantum* antigen, MCH: mean corpuscular hemoglobin, MCV: mean corpuscular volume, MCHC: mean corpuscular hemoglobin concentration, RBC: red blood cells concentration, WBC: leukocytes concentration. * Most platelet results were obtained qualitatively due to platelet aggregation.

**Table 6 animals-13-01649-t006:** Relationship between numerical biochemistry parameters and LeishVet clinical staging of the seropositive sick dogs.

Parameters (Reference Intervals, Units)	Stage II (n = 21) Median (Min–Max)	Stage IIa (n = 59) Median (Min–Max)	Stage IIb (n = 10) Median (Min–Max)	Stage III (n = 16) Median (Min–Max)	*p*-Value (Kruskal-Wallis H Test)
**Total protein (54–71, g/L)**	81.5 (66–117)	74 (59 -103)	68 (60–89)	75.5 (71–110)	0.01697 ^a^
**Albumin (26–33, g/L)**	33 (24–47)	34 (26–56)	31 (26–35)	30 (25–39)	0.03142 ^b^
**Globulin (27–44, g/L)**	45.5 (37–81)	40 (29–71)	36 (30–63)	46 (37–75)	0.001 ^c^
**A/G ratio (0.86–1.93)**	0.68 (0.27–0.89)	0.82 (0.26–1.33)	0.86 (0.35–1.34)	0.52 (0.27–1.52)	0.04389 ^d^
**ALT (21–102, U/L)**	31.5 (12–278)	44 (17–1037)	38 (18–88)	52 (21–228)	0.0239 ^e^
**ALP (20–156, U/L)**	63.5 (20–107)	55 (20–1271)	48.5 (20–105)	51.5 (20–861)	0.71
**Creatinine (0.5–1.5, mg/dL)**	0.94 (0.6–1.22)	0.8 (0.43–1.39)	0.77 (0.62–1.21)	0.81 (0.47–1.9)	0.42
**Urea (21.4–59.9, mg/dL)**	35 (14–54)	32 (14–62)	37.5 (24–80)	46.5 (19–150)	0.046 ^f^
**Serum electrophoresis (g/L)** **Sero-albumin (24.4–49.6)** **Globulins (27–44)** **Alpha-1 globulin (1.7–4.5)** **Alpha-2 globulin (3.8–10.2)** **Beta globulin (8–18)** **Gamma globulin (2.6–11.7)**	31.1 (21.5–38.2) 47.6 (37.8–86) 3.9 (2–8.4) 7.95 (5.2–18.2) 15.8 (12.2–37.5) 19.1 (10.1–60.6)	33.3 (21.2–49.1) 40.1 (28–80.8) 3.6 (2–5.9) 8 (4.26–18.1) 13.7 (9.2–37.9) 14 (5.4–50.8)	31.6 (23–36.6) 36.2 (27.4–66) 3.7 (2.5–4.6) 8.9 (5.4–10.2) 15.5 (9.4–18.3) 10.2 (5.3–34.6)	27.2 (19.1–45.2) 51.1 (29.8–81.5) 3.5 (2.7–5.5) 8.3 (6.5–13.6) 16 (10.3–24.8) 21.9 (8.6–56)	0.0199 ^g^ 0.0006 ^h^ 0.93 0.69 0.32 0.004 ^i^
**UPCR (<0.5)**	1.1 (0.2–1.92)	0.11 (0.02–0.58)	0.96 (0.61–1.84)	1.93 (0.13–101.75)	<0.0001 ^j^
**USG (1006–1050, g/L)**	1033.5 (1026–1041)	1034.5 (1007–1056)	1020.5 (1008–1042)	1024 (1008–1046)	0.0194 ^k^

^a^ χ^2^ = 10.196; ^b^ χ^2^ = 8.845; ^c^ χ^2^ = 16.192; ^e^ χ^2^ = 9.445; ^f^ χ^2^ = 7.994; ^h^ χ^2^ = 17.32; ^i^ χ^2^ = 1.41; ^j^ χ^2^ = 53.9; ^k^ χ^2^ = 11.73. ^d,g^ One-way ANOVA analysis: ^d^ F = 3.822; ^g^ F = 5.5. Stage IV is not included due to a small number of observations (n = 1). Abbreviations: A/G: albumin/globulin ratio, ALP: alkaline phosphatase, ALT: alanine transaminase, max: maximum, min: minimum, UPCR: urinary protein creatinine ratio, USG: urine specific gravity.

**Table 7 animals-13-01649-t007:** Spearman’s correlation (*rs*) between biochemical parameters and age, serology and IFN-γ concentration after LSA or ConA stimulation.

Parameters (Units)	Age		Serology ELISA Units		IFN-γ Concentration (LSA Stimulation, pg/mL)		IFN-γ Concentration (ConA Stimulation, pg/mL)	
	** *r_s_* **	** *p-* ** **Value**	** *r_s_* **	** *p-* ** **Value**	** *r_s_* **	** *p-* ** **Value**	** *r_s_* **	** *p-* ** **Value**
**Total protein (g/L)**	−0.071	0.31	0.4	**<0.0001**	−0.2109	**0.0124**	−0.1507	0.0756
**Albumin (g/L)**	−0.06	0.39	−0.223	**0.001**	0.1381	0.1037	0.1109	0.1921
**Globulin (g/L)**	−0.008	0.91	0.51	**<0.0001**	−0.2593	**0.002**	−0.2206	**0.0088**
**A/G ratio**	−0.01	0.13	−0.55	**<0.0001**	0.2522	**0.0027**	0.2261	**0.0072**
**ALT (U/L)**	0.24	**0.0004**	−0.095	0.18	0.1345	0.1185	0.1032	0.2319
**ALP (U/L)**	0.13	0.07	0.14	**0.041**	−0.05758	0.4992	−0.03891	0.6481
**Creatinine (mg/dL)**	−0.18	**0.007**	−0.079	0.26	−0.01492	0.8611	0.1262	0.1373
**Urea (mg/dL)**	−0.091	0.19	−0.06	0.4	−0.06855	0.4210	0.06555	0.4416
**Serum electrophoresis (g/L)**								
**Albumin**	−0.13	0.06	−0.46	**<0.0001**	0.1792	**0.0342**	0.1899	**0.0246**
**Globulins**	0.032	0.64	0.53	**<0.0001**	−0.2499	**0.0029**	−0.2407	**0.0042**
**Alpha-1 globulin**	−0.03	0.69	0.11	0.12	−0.08755	0.3036	−0.1649	0.0516
**Alpha-2 globulin**	0.13	0.07	0.31	**<0.0001**	−0.2067	**0.0146**	−0.2083	**0.0139**
**Beta globulin**	0.003	0.96	0.16	**0.02**	−0.06619	0.4371	−0.0768	0.3671
**Gamma globulin**	0.05	0.49	0.6	**<0.0001**	−0.2295	**0.0064**	−0.2273	**0.0069**
**UPCR**	0.15	**0.04**	0.2	**0.007**	0.1315	0.2888	−0.08458	0.4962
**USG (g/L)**	−0.075	0.32	−0.054	0.48	−0.0269	0.7715	0.1352	0.1426

Abbreviations: A/G ratio: albumin/globulin ratio, ALP: alkaline phosphatase, ALT: alanine transaminase, ConA: concavalin A, LSA: soluble *L. infantum* antigen, UPCR: urinary protein creatinine ratio, USG: urine specific gravity.

**Table 8 animals-13-01649-t008:** Significant differences between categorical clinicopathological alterations and LeishVet clinical staging.

Categorical Alterations (Number of Dogs)	Stage II (n = 21) N (%)	Stage IIa (n = 59) N (%)	Stage IIb (n = 10) N (%)	Stage III (n = 16) N (%)	Stage IV (n = 1) N (%)	*p*-Value (Chi-Squared; *df*)
**Thrombocytosis (n = 2)**	0 (0)	2 (3.4)	0 (0)	0 (0)	0 (0)	<0.001 (37.8; 8)
**SPE pattern: chronic antigenic stimulation (n = 69)**	15 (75)	40 (67.8)	4 (40)	10 (62.5)	0 (0)	0.004 (22.9; 8)
**Hyperglobulinemia (n = 48)**	12 (60)	21 (35.6)	3 (30)	12 (75)	0 (0)	0.022 (11.5; 4)
**Increased alpha-2-globulins (n = 14)**	3 (15)	5 (8.5)	0 (0)	6 (37.5)	0 (0)	0.025 (11.1; 4)
**Hypergammaglobulinemia (n = 79)**	19 (95)	44 (74.6)	4 (40)	12 (75)	0 (0)	0.009 (13.6; 4)
**Decreased A/G ratio (n = 68)**	16 (80)	33 (55.9)	4 (40)	14 (87.5)	1 (100)	0.029 (10.81; 4)
**Renal azotemia (n = 2)**	0 (0)	0 (0)	0 (0)	1 (6.3)	1 (100)	<0.001 (55.4; 4)
**Increased creatinine (n = 3)**	0 (0)	0 (0)	0 (0)	2 (12.5)	1 (100)	<0.001 (46.4; 8)
**Increased urea (n = 6)**	0 (0)	1 (1.7)	1 (10)	3 (18.8)	1 (100)	0.001 (27.8; 8)
**Proteinuria (n = 31)**	1/2 (50)	4/58 (6.9)	10/10 (100)	15/16 (93.8)	1/1 (100)	<0.001 (64.5; 4)
**Isosthenuria (n = 10)**	0/2 (0)	4/58 (6.9)	2/10 (20)	3/16 (18.8)	1/1 (100)	0.03 (10.7; 4)

Abbreviations: A/G: albumin/globulin, SPE: serum protein electrophoresis.

## Data Availability

Data of the study are available from the corresponding author upon reasonable request.

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
