# Peer review of "Signalment, Immunological and Parasitological Status and Clinicopathological Findings of Leishmania-Seropositive Apparently Healthy Dogs"

_animals, 2023, doi:10.3390/ani13101649_

Round 1

Reviewer 1 Report

The manuscript is well detailed, with a lot of interesting information.

Only minor comments are to be noted:

 - it would have been better to use the term qPCR rather than rt-PCR, however, it is true that the term rt-PCR is used in several other articles

- lines 161 to 166, the authors describe the criteria which determine whether a sample is highly, moderately or weakly positive. The values retained are lower than those proposed in the article on which they are based, cited in reference. What criteria did they use to establish these limits?

- line 179, even if the authors indicate the reference publications, it would be better to indicate the sequence of the primers and probe used and briefly the qPCR conditions

- line 185, the authors consider that a result is positive if the 3 replicates are positive. Aren't they likely to have an underestimation of the positive samples if 2 out of 3 replicates are positive?

- line 229-230, the authors indicate the majority races which include only 16.5% of all races. Are there many other different races?

- in table 1, in the "ELISA positivity", "seropositive healthy", there is a total of 98.1% instead of 100% (less important, 0.1% is missing in "LeishV and stage", "seropositive sick". Line 518 the value 55.1% should be verified).

- line 322, is the value 17.9% correct?

- the manuscript contains a lot of tables, but these help to follow better. It would have been better to indicate the values from lines 335-338, lines 366-389, lines 366-389, lines 403-405, in a table (in supplementary data if necessary).

- lines 390-393, did the evaluation of the data carried out in relation to the reference intervals indicated in the table or in comparison between dogs?

- lines 407-409, did the analyzes carried out from the values indicated in table 8?

Reviewer 2 Report

Please refer to attached Word document.

Please refer to attached Word document.

Reviewer 3 Report

The topic addressed by the authors is very interesting and includes important information on Leishmaniosis, especially subclinical canine Leishmaniosis in the Mediterranean basin.

I have some minor concerns which are as follows:

L 20: “and clinic….”; instead, and, add; correlated with.

L 29: instead, go, add; remain.

L 29: instead, without performing, add; aimed to the lack of.

L 30: after “of these tests” please add, with the combination of clinical signs.

L 45: L 29: instead, abnormalities, please add; findings.

L 53: “is considered the main reservoir of L. infantum infection”; Here could be referred also other reservoirs of the parasite such as wild animals and please add the following reference

Morales-Yuste, M., Martín-Sánchez, J., & Corpas-Lopez, V. (2022). Canine leishmaniasis: Update on epidemiology, diagnosis, treatment, and prevention. Veterinary Sciences, 9(8). doi:10.3390/vetsci9080387

L 66 “non-regenerative anemia, serum proteins imbalances, and proteinuria”; Please rephrase.

L 82; Instead, presentation, please add; profiles or spectrum of …..

L 106; Instead, and, please add; combined thoroughly with the ….

L 231; Instead, between, please add; related to….

L 317; Instead, almost all, please add; the majority of them…

L 331; After (13.2%); please delete the word and.

L 366: Instead, was, please add; were…

L 371: Instead, biochemistry, please add; biochemical…

L 389: Instead, when, please add; compared to…

L 498: Please delete; ….at diagnosis and….

L 542: Instead, disease, please add; disorders…

L 628: At the end of the sentence please add; both in vitro and in vivo therapeutic potential treatments. And also, please add the following reference:

Tzora, A., Lawrence, F., & Robert-Gero, M. (1998). The effect of sinefungin on the macromolecular biosynthesis of Leishmania species. Journal of the Hellenic Veterinary Medical Society, 49(2), 137-142. doi:10.12681/jhvms.15765.

L 639: At the end of the sentence please add; preventing the spread of the diseases that concerns the Public Health. 

Minor editing of English language required
